# The Association of Anti-Inflammatory Diet Ingredients and Lifestyle Exercise with Inflammaging

**DOI:** 10.3390/nu13113696

**Published:** 2021-10-21

**Authors:** Edyta Wawrzyniak-Gramacka, Natalia Hertmanowska, Anna Tylutka, Barbara Morawin, Eryk Wacka, Marzena Gutowicz, Agnieszka Zembron-Lacny

**Affiliations:** Department of Applied and Clinical Physiology, Collegium Medicum University of Zielona Gora, 28 Zyty Str., 65-417 Zielona Gora, Poland; e.gramacka@cm.uz.zgora.pl (E.W.-G.); n.hertmanowska@cm.uz.zgora.pl (N.H.); a.tylutka@cm.uz.zgora.pl (A.T.); b.morawin@cm.uz.zgora.pl (B.M.); e.wacka@cm.uz.zgora.pl (E.W.); m.gutowicz@cm.uz.zgora.pl (M.G.)

**Keywords:** cytokines, nutritional status, physical performance, C-reactive protein, cell-free DNA

## Abstract

One of the latest theories on ageing focuses on immune response, and considers the activation of subclinical and chronic inflammation. The study was designed to explain whether anti-inflammatory diet and lifestyle exercise affect an inflammatory profile in the Polish elderly population. Sixty individuals (80.2 ± 7.9 years) were allocated to a low-grade inflammation (LGI *n* = 33) or high-grade inflammation (HGI *n* = 27) group, based on C-reactive protein concentration (<3 or ≥3 mg/L) as a conventional marker of systemic inflammation. Diet analysis focused on vitamins D, C, E, A, β-carotene, n-3 and n-6 PUFA using single 24-h dietary recall. LGI demonstrated a lower n-6/n-3 PUFA but higher vitamin D intake than HGI. Physical performance based on 6-min walk test (6MWT) classified the elderly as physically inactive, whereby LGI demonstrated a significantly higher gait speed (1.09 ± 0.26 m/s) than HGI (0.72 ± 0.28 m/s). Circulating interleukins IL-1β, IL-6, IL-13, TNFα and cfDNA demonstrated high concentrations in the elderly with low 6MWT, confirming an impairment of physical performance by persistent systemic inflammation. These findings reveal that increased intake of anti-inflammatory diet ingredients and physical activity sustained throughout life attenuate progression of inflammaging in the elderly and indicate potential therapeutic strategies to counteract pathophysiological effects of ageing.

## 1. Introduction

Inflammation is a physiological response to tissue damage and is designed to protect the host from infections by eliminating pathogens, promoting cellular repair and restoring homeostatic conditions [1]. However, the ageing immune system comes with an apparent paradox. Despite a decrease in immune responsiveness to infection and vaccination, and even a reduction in infection-associated immunopathology, the elderly experience a systemic inflammation, which increases the risk of cardiovascular, neurodegenerative and autoimmune diseases and can significantly aggravate health status. With the advance of the process of ageing of the immune system, the elderly also become more susceptible to infectious diseases and cancers [1]. Moreover, the activation of inflammatory pathways, including mammalian target of rapamycin (mTOR) and nuclear factor erythroid-related factor 2 (Nrf-2) signalling, appears to be involved in the pathophysiology of sarcopenia and frailty [2]. One of the most recent theories on ageing focuses on the immune response, and takes into consideration the activation of subclinical, chronic low-grade inflammation. Inflammaging is manifested by the release of a large number of inflammatory mediators that are produced to repair damage at tissue level, such as interleukins IL-1, IL-2, IL-6, IL-8, IL-12, IL-13, IL-15, IL-18, IL-22, IL-23, tumour necrosis factor α (TNFα) interferon-γ (IFN-γ) as pro-inflammatory cytokines, and IL-1Ra, IL-4, IL-10, transforming growth factor (TGF-β1) as anti-inflammatory cytokines, and also lipoxin A4 and heat shock proteins as mediators of cytokines [3,4]. According to Minciullo et al. [3], inflammaging is a key to understanding ageing, and anti-inflammaging may be one of the secrets of longevity. Therefore, it is important to contemplate inflammaging and to intervene more quickly and multidimensionally with preventive and therapeutic approaches [5].

Of late, there has been some interest in the changes to modifiable lifestyle factors, which can significantly attenuate inflammaging [6,7]. The diet ingredients play an important role in the progression of inflammation, with certain foods and nutrients being capable of eliciting immunomodulatory effects. Most human studies have concentrated on analyses of habitual dietary intake and systemic markers of inflammation such as high-sensitivity C-reactive protein (CRP), IL-6 and TNFα, which are also strong predictors of all-cause mortality risk in 80-year-old people [8]. The available evidence indicates that consumption of vegetables and fruit, or macro- and micronutrients, n-3 polyunsaturated fatty acids (PUFA), monounsaturated fatty acids (MUFA), flavonoids, vitamin C and E has been shown to reduce systemic inflammation, whereas saturated fatty acids (SFA), high glycaemic index carbohydrates, and a high dietary n-6 to n-3 PUFA ratio increase serum levels of pro-inflammatory cytokines [9,10]. Healthy eating patterns such as the Mediterranean and vegetarian diets may ameliorate inflammatory processes and decrease the levels of circulating inflammatory biomarkers, thereby reducing the risk of age-related diseases [11,12]. Meta-analyses or systemic reviews of observational studies have reported lower serum concentrations of CRP, IL-6 and TNFα among vegetarian and Mediterranean diet eaters compared with omnivores [12,13]. Regular physical activity including cardiovascular and resistance exercise has been associated with lower levels of inflammatory mediators, mainly CRP, IL-6 and TNFα, as well as higher anti-inflammatory capacity [14,15], improved neutrophil chemotaxis [16], natural killer (NK) cell cytotoxicity and increased T lymphocyte proliferation [17], as well as a stronger post-vaccination response [18]. Daily physical activity, which particularly affects immunity and dramatically declines with age, has not been widely investigated yet. Recently we demonstrated that lifestyle exercise leads to rejuvenation of the immune system by increasing the percentage of naïve T lymphocytes or by reducing the tendency of the inverse CD4/CD8 ratio, which means a low risk of chronic inflammatory diseases in the elderly [19]. Therefore, our present study was designed to assess the intake of anti-inflammatory diet ingredients and to explore the association with the major biomarkers of the ageing immune system in the Polish elderly population. Moreover, we investigated whether CRP, as a conventional marker of systemic inflammation, is useful to differentiate inflammaging and anti-inflammaging in conjunction with a diet and physical activity and, thus, to facilitate therapeutic strategies to counteract the pathophysiological effect of ageing.

## 2. Materials and Methods

### 2.1. Participants

One hundred and seventy-four individuals were recruited from the University of the Third Age in Zielona Gora (Figure 1), which is an organization encouraging the elderly over 65 years of age to stay active by participating in many educational programmes, including arts, classical studies, discussion classes, computer courses, crafts, drama, film/cinema studies, history, languages, literature, music, social sciences, and physical activity. The current health status of the participants was assessed on the basis of medical records at a routine follow-up visit to a primary care physician. The inclusion criteria included the age 65 years or older and the same access to medical healthcare, under the care of the same medical centre. The exclusion criteria, based on the medical interview, included acute infectious and autoimmunological diseases, uncontrolled hypertension and/or diabetes, oncologic diseases, neurodegenerative diseases and cognitive impairment, musculoskeletal disturbances and an implanted pacemaker. Eventually, 60 participants (females *n* = 28, males *n* = 32) aged 80.2 ± 7.9 years were included in the project (Table 1), and they represented the elderly aged ≥ 65 years (*n* = 17; 70.8 ± 2.5 years), the old elderly aged ≥ 75 years (*n* = 25; 79.4 ± 2.8 years) and the oldest old ≥ 85 years (*n* = 18; 90.1 ± 3.0 years) according to WHO age classification [20]. The elderly were allocated to two groups, i.e., low-grade inflammation (LGI *n* = 33) and high-grade inflammation (HGI *n* = 27), based on the measurement of CRP concentration as a conventional marker of systemic inflammation according to reference values (CRP < 3 mg/L or ≥3 mg/L) for the elderly described by Wyczalkowska-Tomasik et al. [21]. Nineteen participants withdrew from the project during the study due to hospitalization and/or the COVID-19 epidemic. The diet analysis included 41 individuals aged 80.5 ± 8.0 years (LGI *n* = 26, HGI *n* = 15). The medications taken by the participants included antihypertensive (84%) and hypolipidemic (10%) drugs as well as anticoagulants including anti-platelet agents (15%). All the subjects were informed of the aim of the study and signed a written consent to participate in the project. The study protocol was approved by the Regional Bioethics Commissions (Regional Medical Chamber in Zielona Gora, No. 04/133/2020, University of Zielona Gora No. UZ/19/2021), in accordance with the Helsinki Declaration.

### 2.2. Body Composition

Body mass and body composition fat-free mass (FFM) and fat mass (FM) were evaluated by a bioelectrical impedance method using a Tanita Body Composition Analyser MC-980 (Tokyo, Japan) calibrated prior to each test session in accordance with the manufacturer’s guidelines. Duplicate measurements were made in the study participants standing upright, and the average value was included for the final analysis. The measurements were taken between 7:00 and 9:00 a.m., prior to blood sampling, and the recurrence of measurement was 98%. The subjects of the study were advised to fast for at least 12 h before the measurement of body composition. They were asked to avoid heavy physical exertion and to avoid consuming drinks for 3 h before the test. The measurements were made in accordance with the procedure previously used in the elderly [22].

### 2.3. Diet Analysis

In the quantitative assessment of the daily food rations, a single 24-h dietary recall was applied. In order to elaborate the nutritional value of daily food rations, we used the Aliant Soft 2.0 (Gdańsk, Poland), focusing on anti-inflammatory diet ingredients such as vitamins D, C, E, A and β-carotene as well as n-3 PUFA and n-6 PUFA. To evaluate compliance with the recommended dietary intake, the supply of different nutrients was categorized as intake in accordance with the nutritional standards for Polish people [23]. Based on the collected data, the proportion of subjects achieving recommended nutrient intakes was calculated in reference to the recommended dietary allowance (RDA) values, and for some nutrients—adequate intake (AI) values, and the World Health Organisation (WHO) recommendations [24] for people with low physical activity aged over 60. According to Różańska et al. [25], the glycaemic load of daily food ration was considered low for values of 80 g or below, medium for values between 80–120 g and high for values of 120 g or above.

### 2.4. Physical Performance

The 6-min walk test (6MWT) was performed according to technical standards of the European Respiratory Society and American Thoracic Society [26]. The total distance walked in the test was recorded, and the 6MWT gait speed was then calculated by the following equation: 6MWT gait speed (m/s) = total distance (m)/360 s [27]. Following the classification by Middelton et al. [28], a gait speed within the range of 1.0 to 1.3 m/s classified the elderly as active while a gait speed < 1.0 m/s classified them as inactive.

### 2.5. Blood Sampling

Fasting blood samples were collected from the median cubital vein in the morning between 8:00 and 10:00 a.m. using S-Monovette tubes (Sarstedt AG & Co. KG, Nümbrecht, Germany). The whole blood samples were placed into specimen tubes containing EDTA and were immediately analysed. For the other biochemical analyses, blood samples were centrifuged at 3000 rpm for 10 min, and aliquots of serum were stored at −80 °C. The average intra-assay coefficients of variation (intra-assay CV) for the used ELISA kits were <10%. All samples were analysed in duplicate or triplicate in a single assay to avoid inter-assay variability.

### 2.6. Haematological Variables

Peripheral blood morphology including leucocytes, granulocytes (GRA), lymphocytes (LYM), red blood cell count (RBC), haemoglobin (HB), haematocrit (HCT) and platelets (PLT), was determined using 3 diff BM HEM3 Biomaxima (Lublin, Poland).

### 2.7. Biochemical Variables

Serum triglycerides (TG), total cholesterol (TC), high-density lipoproteins (HDL) and low-density lipoproteins (LDL) were determined using BM200 Biomaxima (Poland). The non-HDL cholesterol was calculated by subtracting HDL from total cholesterol concentration. Oxidised low-density lipoprotein (oxLDL) was determined using ELISA kits from SunRed Biotechnology Company (Shanghai, China) with detection limit at 3.03 mg/dL. Glucose, bilirubin and albumin were determined using BM200 Biomaxima (Poland), and lactate was measured using the DP 310 Vario II mobile spectrophotometer Diaglobal (Berlin, Germany).

### 2.8. Inflammatory Variables

Serum C-reactive protein (CRP) was measured using a high sensitivity commercial ELISA kit from DRG International (Springfield Township, Cincinnati, OH, USA) with the detection limit of 0.001 mg/L. The C-reactive protein to albumin ratio (CRP/albumin) was calculated as CRP (mg/L) divided by albumin level (g/L) [29]. Interleukin 1β (IL-1β), interleukin 6 (IL-6), interleukin 8 (IL-8), interleukin 10 (IL-10), interleukin 13 (IL-13) and tumour necrosis factor α concentrations, as markers of inflammaging and anti-inflammaging systems, were determined using ELISA kits from SunRed Biotechnology Company (Shanghai, China) with detection limits of 28.384 pg/mL, 1.867 pg/mL, 0.953 ng/mL, 1.142 pg/mL, 0.413 pg/mL and 2.782 pg/mL, respectively. The total circulating fragments of DNA (cell-free DNA) were measured directly in serum using a Quant-iTTM DNA high-sensitivity assay kit and a Qubit fluorometer (Invitrogen, Carlsbad, CA, USA) in accordance with the manufacturer’s instructions. The samples were analysed in duplicate, and the mean of the two measurements was used as the final value. The intra-assay CV for the Quant-iTTM DNA high-sensitivity assay was <2%.

### 2.9. Statistical Analysis

Statistical analyses were performed using the R version 4.0.3 [30]. The assumptions for the use of parametric or nonparametric tests were checked using the Shapiro-Wilk and Levene’s tests to evaluate the normality of the distributions and the homogeneity of variances, respectively. The significant differences in mean values between the groups were assessed by one-way ANOVA. If the normality and homogeneity assumptions were violated, the Mann-Whitney nonparametric test was used. Additionally, eta-squared (*η*^2^) based on the EtaSq function from DescTools R package was used to measure the effect size that is indicated as having no effect if 0 ≤ *η*^2^ < 0.01, a small effect if 0.01 ≤ *η*^2^ < 0.06, a moderate effect if 0.06 ≤ *η*^2^ < 0.14, and a large effect if *η*^2^ ≥ 0.14. Spearman’s rank correlation (r_s_—Spearman’s rank correlation coefficient) was used to investigate the relationships between inflammatory markers. Statistical significance was set at *p* < 0.05.

## 3. Results

### 3.1. Body Composition

The body mass index (BMI) ranged from 19.9 to 40.8 kg/m^2^. Approximately 26% of the study seniors had normal body mass (18.5–24.9 kg/m^2^), 40% were classified as overweight (25–29.9 kg/m^2^) and 34% as obese (≥30 kg/m^2^). High fat mass dominated among women whereas high fat-free mass was characteristic of men (Table 1). There were no significant differences in components of body composition between LGI and HGI groups (Table 2). Fat mass and fat-free mass were not related to the levels of circulating inflammaging biomarkers.

### 3.2. Diet Analysis

According to the standard of nutrition for the Polish elderly population [23], our results demonstrated some differences in intake of the major ingredients that might influence the rate of inflammaging in LGI and HGI (Table 3). The daily intake of total carbohydrates as well as the values of glycaemic index and glycaemic load did not significantly differ between the LGI and HGI groups. However, HGI tended to have a low intake of n-3 PUFA, especially EPA and DHA, and high intake of n-6 PUFA such as linoleic acid, which resulted in a significant increase in the n-6/n-3 PUFA ratio compared to that in the LGI group. HGI also demonstrated a low percentage of total energy derived from PUFA and from SFA. Furthermore, the diet in the HGI group had a significantly higher total protein content (17.09 ± 3.72% of energy) compared to the diet in the LGI group (14.01 ± 4.19% of energy). The micronutrient intake levels, including vitamins D, C, E, A, β-carotene and magnesium, showed varying levels in the elderly. Daily intake of vitamin D tended to reach higher values in the LGI group with a low inflammatory state (CRP < 3 mg/L). Still, micronutrient intake in the average diet was very low in all participants, particularly in regard to vitamin D and magnesium.

### 3.3. Physical Performance

On average, the participants covered the distance of 330 ± 118 m in 6MWT at a gait speed of 0.92 ± 0.32 m/s. Approx. 23% of the subjects achieved a gait speed below the reference value of 0.8 m/s, which determines the cut-off point for sarcopenia according to Cruz-Jentoft et al. [31]. Moreover, the HGI group with CRP concentration above normal ranges demonstrated a significantly lower gait speed (0.72 ± 0.28 m/s) compared to the LGI group (1.09 ± 0.26 m/s). The value *η*^2^ indicated a large effect of inflammation on gait speed (Figure 2). Other pro-inflammatory mediators, i.e., IL-1β, IL-6, IL-13 and TNFα, also demonstrated high concentrations in the elderly with low gait speed.

### 3.4. Haematological Variables

The white blood cell count fell within the referential range in all participants, whereas higher granulocytes and GRA% were detected in the HGI compared to the LGI group (Table 4). The value *η*^2^ indicated a large effect of inflammation on white cell counts. The parameters of the red blood cells, such as RBC, HB and HCT, were below the referential values in approx. 50% of our participants. However, *η*^2^ analysis showed no inflammation effect.

### 3.5. Biochemical Variables

Total cholesterol and lipoproteins have been proven to be the strongest biomarkers of ageing and nutritional status [32]. High levels of TG > 150 mg/dL, TC > 200 mg/dL, LDL > 130 mg/dL and non-HDL > 130 mg/dL were found in 25% of the study’s old elderly. However, the lipoprotein-lipid profile, including oxLDL, did not differ between the LGI and HGI groups (Table 5). Similarly, an elevated glucose level is known as a biomarker of ageing and nutritional status, and it is associated with alterations in metabolic and hormonal function, including altered expression of cellular insulin receptors and glucose transporter units in target tissues. Some participants (*n* = 6) demonstrated an increased glucose level > 115 mg/dL without being diagnosed with diabetes. Serum lactate and bilirubin levels, as biomarkers of age-related diseases, were within normal ranges; however, bilirubin was significantly increased in the HGI group. The value of *η*^2^ indicated a moderate effect of inflammaging on elevated bilirubin concentration, which may be associated with increased mortality in the elderly [33]. Albumin, as an indicator of malnutrition in the elderly in clinically stable conditions, was recorded within normal ranges. However, the HGI group demonstrated a significantly lower concentration of albumin compared to the LGI group. Serum albumin was reported to decrease with increasing age by approx. 0.1 g/L per year, with the main reason being high concentrations of IL-6 and TNFα [34].

### 3.6. Inflammatory Variables

Most observational studies and clinical trials have used high-sensitivity CRP as a biochemical marker of inflammation because it is relatively stable and easy to measure. In our study, CRP concentration was found to be within normal ranges in 33 individuals and above normal ranges in 27 individuals (Table 6). The ratio of CRP to albumin was 5-fold higher in HGI than in LGI (Figure 3) and highly correlated with cfDNA (Figure 4) as a biomarker of age-associated inflammation and frailty. cfDNA highly correlated with other markers of inflammaging (Table 7). Classical cytokine biomarkers of ageing including IL-1β, IL-6 and TNFα were higher in the HGI group (Table 6), which also demonstrated a low level of physical activity and poorer adherence to the recommended intake of PUFA. The value *η*^2^ indicated a particularly large effect of inflammaging on IL-6, which is called “a cytokine for gerontologists” [3]. Furthermore, elevated concentrations of pro-inflammatory IL-13 were also significantly related to inflammaging, in contrast to IL-8, which was recorded at similar levels in the LGI and HGI groups. IL-10 is arguably the most potent anti-inflammatory factor, and its significant increase was observed in the individuals with low CRP concentrations. The value *η*^2^ indicated a large effect of inflammaging on the IL-10 level.

## 4. Discussion

Nutritional deficiencies and low physical activity, which are common among people aged over 65 years, cause an essential loss of physiologic reserves, making the individual susceptible to the activity of inflammatory mediators [35]. Earlier, we investigated the relationship between lifestyle exercise and percentage of CD4+ and CD8+ naïve and memory T lymphocytes as well as CD4/CD8 ratio in active compared to inactive older individuals [19]. Furthermore, we also analysed changes in inflammatory pattern in response to physical exercise intervention [22]. In the present study, we focused on the profile of circulating pro- and anti-inflammatory mediators in conjunction with nutritional status and physical performance resulting from daily activity. The diet analysis showed nutritional frailty in the intake of the major ingredients, which may influence the rate of inflammaging. In both analysed groups, LGI and HGI, poor compliance with the recommended dietary intake nutrients and food components was observed. The HGI group demonstrated a higher daily intake of proteins but a lower intake of vitamin D. This was related to a higher consumption of pro-inflammatory components (red meat, processed meat, eggs, sugar-sweetened beverages and refined grains, saturated fat and sweets) and a low consumption of anti-inflammatory components (leafy green vegetables, dark yellow vegetables, fruit juice, oily fish). Moreover, the elderly with CRP ≥ 3 mg/L showed a lower adherence to the recommended intake of essential fatty acids, such as EPA and DHA. The available data indicate the beneficial effect of PUFA, with the emphasis placed especially on n-3 essential fatty acids as the greatest anti-inflammatory ingredients [36]. Galland [9] demonstrated that both a high intake of n-3 PUFA and plasma levels of total n-3 PUFA and EPA are inversely associated with CRP, IL-6 and TNFα. Ferucci et al. [37] observed that both EPA and DHA demonstrated anti-inflammatory effects and were inversely associated with IL-6 and TNFα but positively associated with IL-10 concentrations. Recently, Felix-Soriano et al. [38] demonstrated that supplementation with a DHA-rich fish oil concentrate had beneficial effects on cardiometabolic health markers in postmenopausal women. In our study, the adherence to dietary recommendations concerning PUFA was better in the diet of the elderly in the LGI group compared to those in the HGI group. The dietary content of n-6 PUFAs such as linoleic and arachidonic acid were predominant in the diet of the HGI group. There is evidence that a high n-6 fatty acid diet inhibits the anti-inflammatory and inflammation-resolving effect of the n-3 fatty acids [39]. Vitamin D has the potential to reduce inflammatory state. Many studies have consistently found little, if any, association between vitamin D status and circulating markers of inflammation [36]. Significant associations between low vitamin D status and CRP, IL-6 and IL-10 ratio have been reported in 957 adults from Northern Ireland. The subjects defined as vitamin D deficient were significantly more likely to have an IL-6 to IL-10 ratio > 2 compared with those defined as sufficient [40]. In the HGI group, the ratio IL-6 to IL-10 was approx. > 1.3, whereas in the LGI group, the ratio was found to be <1. Nonetheless, the elderly are often vitamin D deficient, and there is a rationale for using vitamin D as an anti-inflammatory agent. Randomized trials are still necessary to provide strong clinical evidence and a dose-response relationship with an inflammatory response. Interestingly, the total protein intake was significantly higher in the HGI than the LGI group. According to the standard of nutrition for the Polish elderly population [23], daily protein intake should provide 15–20% of total energy to prevent deficiencies and achieve an optimal intake to maintain health and functioning. On the other hand, high protein intake accelerates the decline in renal function and enhances the incidence of inflammatory bowel disease specific to ingestion of animal protein [41]. The role of increasing protein content in the diet and the influence of this modification on inflammation is controversial. There is no literature regarding the influence of protein intake on inflammation, and whether varying protein content in the diet may have some beneficial effects in that respect remains elusive [42]. The Metabolic Syndrome Reduction in Navarra (RESMENA) project revealed a direct relationship between dietary protein intake and inflammation. Individuals who consumed at least 67.1 g of protein had a higher inflammatory score than participants who consumed less protein. Animal protein has been shown to have an impact on the inflammatory score, whereas vegetable protein intake did not produce such an effect [42]. We may conclude that it is not the overall quantity of protein intake, but the source of the dietary protein that is crucial in relation to an anti-inflammatory diet. Ricker and Haas indicated that protein in an anti-inflammatory diet should be primarily plant-based, including legumes, soy, nuts and seeds, with some sources in fish and small amounts of lean meat. Animal protein contains higher levels of n-6 PUFA, and an anti-inflammatory diet should include protein sources containing higher levels of n-3 PUFA [43].

Most observational studies and clinical trials have used CRP as a biochemical marker of systemic inflammation that is produced following stimulation by various cytokines that can be drivers of an acute response to infection, ischemia, trauma, and other inflammatory conditions such as physical inactivity [44,45,46]. Contrary to CRP, albumin is reduced in the chronic inflammatory process; therefore, the CRP to albumin ratio can be a more sensitive marker to predict inflammation than CRP or albumin rates alone, due to their two different directions (increased CRP and decreased albumin) [47]. Systemic inflammation, and a high concentration of IL-6 and TNFα in particular, not only reduces albumin synthesis but also increases its degradation and promotes its transcapillary leakage. Decreased plasma albumin has been considered as a risk factor for nutritional deficiencies and frailty in the elderly [48,49]. Onem et al. [49] showed the relationships between albumin concentration and cognitive and motor functions in patients from nursing homes. They concluded that restoring the reference values of albumin could improve cognitive and motor functions in the elderly. Based on this knowledge, we point to the CRP/albumin ratio as a useful marker of inflammaging in relation to nutritional status and physical performance. So far, there has been no report regarding the prognostic significance of CRP/albumin in ageing. We demonstrated for the first time that nutritional frailty and slow gait speed (<0.8 m/s) were closely related to high CRP/albumin ratio. The ratio of CRP/albumin was <0.07 in the LGI group and >0.07 in the HGI group. The highest values > 0.2 were observed in 26% of participants in the HGI group, which predicts poor prognosis in the elderly according to Sun et al. [50], who observed that cancer patients with a CRP/albumin ratio ≥ 0.189 ran a greater risk of mortality. Nevertheless, further studies are still needed to explain the epidemiological and clinical significance of the CRP/albumin ratio in predicting the overall survival of the elderly in conjunction with lifestyle elements.

Chronic inflammatory state was also manifested by the release of large amounts of pro-inflammatory IL-1β, IL-6, IL-13 and TNFα from immune cells, and peripheral tissues have been shown to play an important role in the pathogenesis and progression of inflammaging [3]. High levels of IL-1β, together with IL-6 and TNFα, are associated with an increased risk of morbidity and mortality in the elderly. In particular, cohort studies have indicated that IL-1β, IL-6 and TNFα are involved in the alteration of nutritional status, poor physical performance, loss of muscle strength, cognitive decline, and cardiological, neurological and vascular events [51]. IL-6 expression is normally low in the absence of inflammation, but its elevated serum level is characteristic of ageing [52] and may reflect age-related pathological processes that develop over decades, even in apparently healthy subjects [53,54]. The study by Ma et al. [55] showed that after adjustment for age and sex, circulating IL-6 level correlated negatively with exercise tolerance in the elderly. IL-13 and TNFα were also detected at elevated levels in most studies of elderly populations and were associated with reduced functional capacity and frailty and increased mortality [3,4,56]. In our study, higher levels of pro-inflammatory cytokines IL-1β, IL-6, IL-13 and TNFα were clearly associated with nutritional frailty and poor physical performance in the HGI group in contrast to anti-inflammaging status confirmed by significantly elevated levels of IL-10 in the LGI group. IL-10 inhibits the production of IL-1β, IL-6, IL-8 and TNFα and plays an important role in orchestrating the inflammatory reaction involving the activation of neutrophils, monocytes/macrophages, natural killer cells and T and B cells and in their recruitment to the sites of inflammation [57]. The most elderly from the LGI group with a high concentration of IL-10 demonstrated a higher gait speed (≥1.0 m/s) and higher intake of anti-inflammatory ingredients such as n-3 PUFA and vitamin D, despite the fact that all participants did not reach the recommended intake of essential fatty acids and micronutrients.

Studies in the last decade have indicated cfDNA as a potent biomarker that could provide important insight into the pathogenesis of many age-related diseases [58,59,60]. The release of cfDNA into the circulation is proportional to the severity of the systemic inflammation, and the cellular source of cfDNA is often neutrophil extracellular trap formation. Widespread cell death of lymphocytes and endothelial cells as well as an organ dysfunction characteristic of ageing may also contribute to extracellular release of DNA [61]. Jylhävä et al. [62] introduced cfDNA into the field of ageing, suggesting that it serves as a novel biomarker of inflammaging. Furthermore, they identified the relationships between the cfDNA species (methylated vs. unmethylated cfDNA) and age-associated inflammation, immunosenescence and frailty [58]. Teo et al. [59] suggested that cfDNA profiling could be used not only as a biomarker of age but also as a predictor of healthy status. We observed that 26% of participants from the HGI group demonstrated a very high concentration of cfDNA above 1000 ng/mL, which may aggravate immunoinflammatory reactivity according to Jylhävä et al. [58]. The level of cfDNA highly correlated with other age-associated inflammation mediators such as CRP, IL-1β, IL-10 and TNFα. This means that plasma/serum cfDNA quantification may have some diagnostic value to assess the impact of lifestyle factors that significantly attenuate inflammaging.

## 5. Conclusions

This study generally supports the notion that anti-inflammatory diet ingredients and physical activity sustained throughout life are critical for optimal inflammatory response in the elderly. Moreover, it shows that the analysis of inflammatory profile, including novel inflammatory markers such as CRP/albumin and cfDNA, with nutritional status and physical performance may be useful in defining healthy or unhealthy ageing (Figure 5). However, future studies are needed to determine the effectiveness of, and conditions for, various nutritional and physical intervention regimens to improve the function of the ageing immune system.

## 6. Limitations

Several limitations of the analysis should be considered. Firstly, the study included a relatively small number of participants, confined predominantly to a population belonging to the same geographic location and having mostly similar lifestyles. Secondly, the lack of information on the participants’ exposure to pathogens throughout life may also have disproportionately affected the analysed markers of inflammaging. Thirdly, the repeated 24-h dietary recall from non-consecutive days might be a better approach to data collection in a large-scale project.

## Figures and Tables

**Figure 1 nutrients-13-03696-f001:**
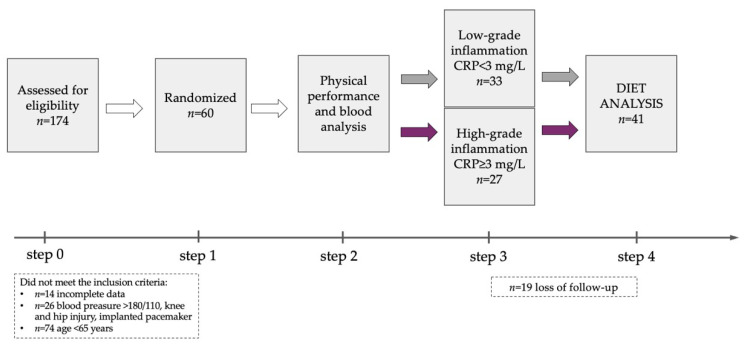
Study flow diagram.

**Figure 2 nutrients-13-03696-f002:**
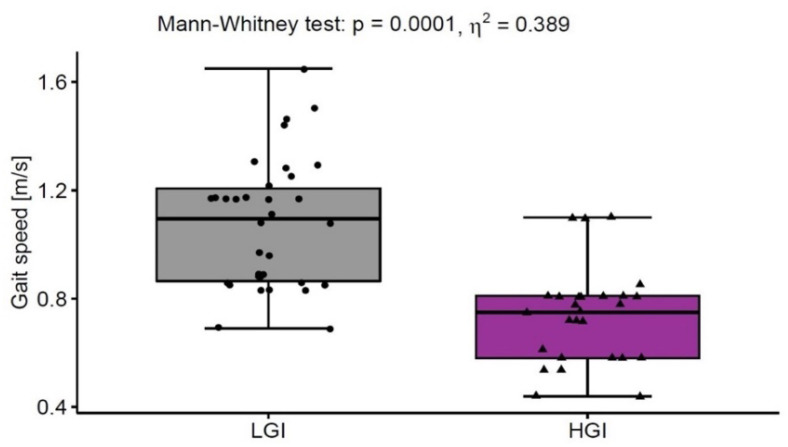
The gait speed in low-grade inflammation group (LGI, *n* = 33) with normal ranges of CRP (<3 mg/L) and in high-grade inflammation group (HGI, *n* = 27) with above normal ranges of CRP (≥3 mg/L).

**Figure 3 nutrients-13-03696-f003:**
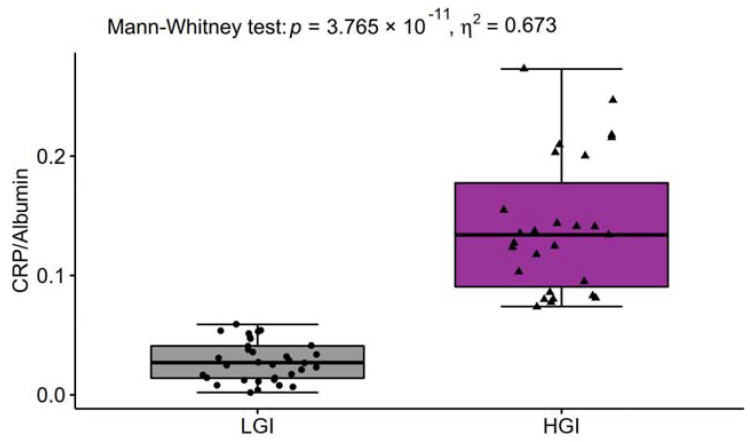
The ratio of C-reactive protein (CRP) to albumin in low-grade inflammation group (LGI, *n* = 33) and in high-grade inflammation group (HGI, *n* = 27).

**Figure 4 nutrients-13-03696-f004:**
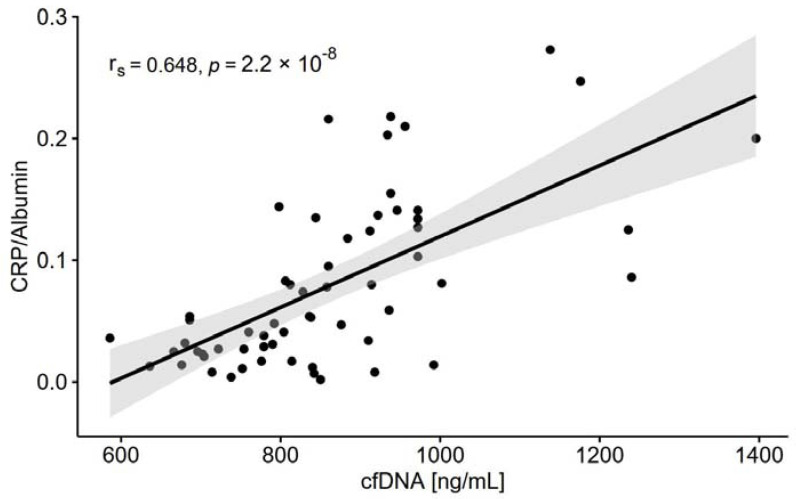
The relationship between cell-free DNA (cfDNA) and C-reactive protein (CRP) to albumin ratio (*n* = 60); r_s_, Spearman’s rank correlation coefficient.

**Figure 5 nutrients-13-03696-f005:**
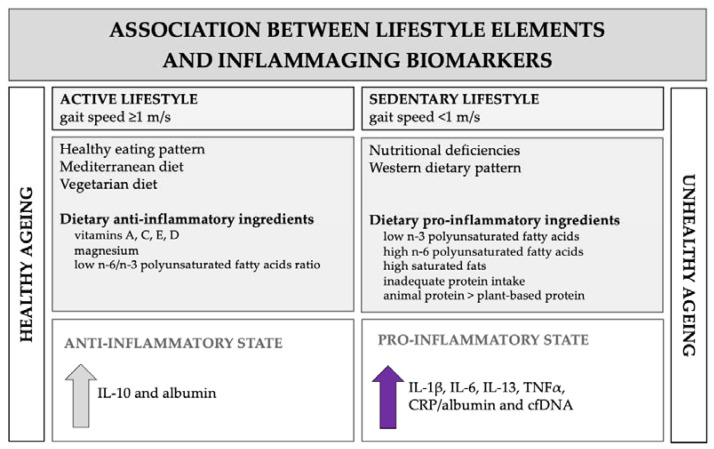
Association between dietary intake, physical activity and biomarkers of the ageing immune system.

**Table 1 nutrients-13-03696-t001:** Anthropometrics, body composition and physical performance (mean ± SD).

	Females *n* = 28	Males *n* = 32	Females vs. Males *p* Level
Age [year]	79.8 ± 8.4	80.7 ± 8.0	0.752
Weight [kg]	67.6 ± 9.7	78.1 ± 14.2	<0.05
Height [cm]	155.5 ± 6.2	166.2 ± 8.2	<0.001
BMI [kg/m^2^]	27.4 ± 4.2	28.4 ± 4.7	0.506
FM [kg]	22.6 ± 7.4	19.7 ± 7.8	<0.05
FM%	31.1 ± 9.0	23.9 ± 6.6	<0.05
FFM [kg]	44.2 ± 5.3	59.1 ± 8.1	<0.001
SBP [mmHg]	140 ± 23	147 ± 20	0.383
DBP [mmHg]	72 ± 10	80 ± 14	0.056
6MWT [m]	279 ± 121	368 ± 104	<0.05
Gait speed [m/s]	0.78 ± 0.34	1.02 ± 0.29	<0.05

Abbreviations: BMI, body mass index; FM, fat mass; FFM, fat-free mass; SBP, systolic blood pressure; DBP, diastolic blood pressure; 6MWT, 6-min walk test.

**Table 2 nutrients-13-03696-t002:** Anthropometrics and body composition in low-grade inflammation (LGI) group and in high-grade inflammation (HGI) group (mean ± SD).

	LGI *n* = 33	HGI *n* = 27	LGI vs. HGI *p* Level
Age [year]	80.9 ± 7.4	79.3 ± 8.3	0.458
Weight [kg]	69.4 ± 11.4	80.3 ± 13.2	*p* < 0.05
Height [cm]	159.8 ± 8.2	164.3 ± 9.4	0.146
BMI [kg/m^2^]	26.8 ± 4.4	29.9 ± 3.7	*p* < 0.05
FM [kg]	19.6 ± 7.5	23.3 ± 7.1	0.096
FM%	26.3 ± 8.8	28.6 ± 7.3	0.416
FFM [kg]	49.4 ± 8.5	57.7 ± 10.4	0.277

Abbreviations: BMI, body mass index; FM, fat mass; FFM, fat-free mass.

**Table 3 nutrients-13-03696-t003:** Daily energy, vitamin and mineral intakes as well as recommended amounts for the Polish elderly population (mean ± SD).

	Recommended Intake [23]	LGI *n* = 26	HGI *n* = 15	LGI vs. HGI *p* Level
Total fat [% of energy]	20–35	34.95 ± 9.24	32.56 ± 7.07	0.393
SFA [% of energy]	<6	14.99 ± 5.83	13.49 ± 4.91	0.405
MUFA [% of energy]	<20	11.49 ± 4.41	11.64 ± 3.22	0.908
PUFA [% of energy]	6–10	5.17 ± 2.56	4.63 ± 1.55	0.466
Linoleic acid (C18:2) [mg]	n/a	7950 ± 4560	9030 ± 4060	<0.05
Arachidonic acid (C20:4) [mg]	n/a	70 ± 50	130 ± 140	0.527
α-Linoleic acid (C18:3) [mg]	n/a	1300 ± 940	1740 ± 1190	0.111
EPA and DHA [mg]	250	260 ± 560	90 ± 120	0.487
n-6 PUFA [g]	n/a	8.03 ± 4.56	9.17 ± 4.06	0.428
n-3 PUFA [g]	n/a	2.09 ± 1.08	1.61 ± 0.92	0.075
n-6/n-3 PUFA [g]	<5:1	4.80 ± 4.22	6.42 ± 2.44	<0.05
Total protein [% of energy]	15–20	14.01 ± 4.19	17.09 ± 3.72	< 0.05
Total carbohydrates [% of energy]	45–65	50.05 ± 7.74	48.89 ± 8.03	0.653
Dietary fibre [g]	18–38	18.35 ± 8.07	21.69 ± 7.85	0.206
Glycaemic index	low < 55 medium 55–70 high > 70	56.21 ± 6.38	59.91 ± 5.87	0.073
Glycaemic load [g]	low ≤ 80 medium 81–120 high ≥ 120	119 ± 56	142 ± 49	0.188
Vitamin D [μg]	15	4.77 ± 7.43	2.41 ± 1.60	0.234
Vitamin C [mg]	F 75 M 90	94.63 ± 61.02	97.15 ± 54.22	0.895
Vitamin E [mg]	F 8 M 10	8.24 ± 4.80	9.59 ± 5.02	0.396
Vitamin A [μg RAE]	F 700 M 900	1100 ± 951	1241 ± 1031	0.660
β-carotene [μg]	n/a	3098 ± 4494	3107 ± 3970	0.995
Magnesium [mg]	F 320 M 420	235 ± 87	293 ± 119	0.082

Abbreviations: LGI, low-grade inflammation group; HGI, high-grade inflammation group; SFA, total saturated fats; MUFA, total monounsaturated fats; PUFA, total polyunsaturated fats; EPA, eicosapentaenoic acid; DHA, docosahexaenoic acid; RAE, retinol activity equivalent; n/a, not applicable; F, female; M, male.

**Table 4 nutrients-13-03696-t004:** Haematological variables (mean ± SD).

	Reference Values	LGI *n* = 33	HGI *n* = 27	LGI vs. HGI *p* Level	*η* ^2^
Leucocytes [10^3^/µL]	5.0–11.6	6.19 ± 1.65	7.83 ± 2.54	<0.001	0.219
Lymphocytes [10^3^/µL]	1.3–4.0	1.64 ± 0.66	1.52 ± 0.71	0.883	0.001
Granulocytes [10^3^/µL]	2.4–7.6	4.26 ± 1.27	5.88 ± 2.10	<0.001	0.371
LYM%	19.1–48.5	28.46 ± 11.46	20.08 ± 8.05	<0.05	0.150
GRA%	43.6–73.4	67.98 ± 8.53	72.42 ± 10.33	<0.001	0.200
RBC [10^3^/µL]	F 4.0–5.5 M 4.5–6.6	4.19 ± 0.67	4.31 ± 0.98	0.841	0.001
HB [g/dL]	F 12.5–16.0 M 13.5–18.0	12.13 ± 1.65	12.60 ± 1.96	0.474	0.010
HCT%	F 37–47 M 40.0–51.0	33.99 ± 4.19	35.99 ± 5.69	0.572	0.006
PLT [10^3^/µL]	150–400	244 ± 74	230 ± 87	0.456	0.011

Abbreviations: LGI, low-grade inflammation group; HGI, high-grade inflammation group; LYM, lymphocytes; GRA, granulocytes; RBC, red blood cells; HB, haemoglobin; HCT, haematocrit; PLT, platelets; F, female; M, male.

**Table 5 nutrients-13-03696-t005:** Lipoprotein-lipid profile and other biochemical variables (mean ± SD).

	Reference Values	LGI *n* = 33	HGI *n* = 27	LGI vs. HGI *p* Level	*η* ^2^
TG [mg/dL]	<150	116 ± 63	134 ± 70	0.165	0.033
TC [mg/dL]	<200	185 ± 48	184 ± 49	0.735	0.002
LDL [mg/dL]	<130	105 ± 44	106 ± 41	0.940	0.001
HDL [mg/dL]	desirable > 60	57.26 ± 14.95	54.00 ± 13.22	0.094	0.048
non-HDL [mg/dL]	<130	128 ± 50	133 ± 41	0.211	0.027
oxLDL [mg/dL]	-	199.74 ± 65.65	173.43 ± 50.55	0.821	0.003
Glucose [mg/dL]	60–115	94.50 ± 16.45	98.91 ± 13.55	0.205	0.028
Lactate [mmol/L]	<2.2	2.75 ± 0.78	2.96 ± 0.72	0.066	0.056
Bilirubin [mg/dL]	<1.0	0.31 ± 0.11	0.41 ± 0.16	<0.05	0.125
Albumin [g/L]	F 37–53 M 42–55	45.90 ± 3.55	43.81 ± 3.49	<0.05	0.152

Abbreviations: LGI, low-grade inflammation group; HGI, high-grade inflammation group; TG, triglycerides; TC, total cholesterol; LDL, low-density lipoproteins; HDL, high-density lipoproteins (non-HDL cholesterol calculated by subtracting the HDL value from TC); oxLDL, oxidized low-density lipoprotein; F, female; M, male.

**Table 6 nutrients-13-03696-t006:** Inflammatory variables (mean ± SD).

	LGI *n* = 33	HGI *n* = 27	LGI vs. HGI *p* Level	*η* ^2^
CRP [mg/L]	1.28 ± 0.75	6.07 ± 2.25	<0.0001	0.695
cfDNA [ng/mL]	747 ± 93	966 ± 148	<0.001	0.381
IL-1β [pg/mL]	1215 ± 493	1459 ± 562	0.068	0.018
IL-6 [pg/mL]	48.15 ± 15.32	62.12 ± 22.82	<0.05	0.178
IL-8 [ng/mL]	8.69 ± 4.68	8.35 ± 3.57	0.486	0.007
IL-10 [pg/mL]	51.42 ± 2.81	48.75 ± 2.95	<0.001	0.281
IL-13 [pg/mL]	2.39 ± 1.44	3.21 ± 1.34	<0.05	0.211
TNFα [pg/mL]	63.97 ± 27.54	82.27 ± 29.83	<0.05	0.075

Abbreviations: LGI, low-grade inflammation group; HGI, high-grade inflammation group; CRP, C-reactive protein; cfDNA, cell-free deoxyribonucleic acid; IL-1β, interleukin 1β; IL-6, interleukin 6; IL-8, interleukin 8; IL-10, interleukin 10; IL-13, interleukin 13; TNFα, tumour necrosis factor α.

**Table 7 nutrients-13-03696-t007:** The relationships of cell-free DNA with other age-associated inflammation markers (*n* = 60). Statistically significant values are marked in bold.

	CRP [mg/L]	IL-1β [pg/mL]	IL-6 [pg/mL]	IL-8 [ng/mL]	IL-10 [pg/mL]	IL-13 [pg/mL]	TNFα [pg/mL]	Bilirubin [mg/dL]	Albumin [g/L]
cfDNA [ng/mL]	**r_s_ = 0.646** *p* < 0.0001	**r_s_ = 0.326** *p* < 0.05	r_s_ = 0.172 *p* = 0.188	r_s_ = 0.020 *p* = 0.882	**r_s_= −0.421** *p* < 0.0001	r_s_ = 0.141 *p* = 0.282	**r_s_ = 0.341** *p* < 0.01	**r_s_ = 0.361** *p* < 0.01	**r_s_= −0.351** *p* < 0.01

Abbreviations: CRP, C-reactive protein; IL-1β, interleukin 1β; IL-6, interleukin 6; IL-8, interleukin 8; IL-10, interleukin 10; IL-13, interleukin 13; TNFα, tumour necrosis factor α; r_s_, Spearman’s rank correlation coefficient.

## Data Availability

The data presented in this study are available on request from the corresponding author.

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
