# Peer review of "The Association of Anti-Inflammatory Diet Ingredients and Lifestyle Exercise with Inflammaging"

_nutrients, 2021, doi:10.3390/nu13113696_

Round 1

Reviewer 1 Report

Although the authors implemented some of the suggested edits, the authors still need to address some of these questions.

1.) The authors still need to update the abstract and introduction section and mention the studies were carried out in a subset of population belonging to the same geographic location and having mostly similar lifestyle and diet. 

2.) The authors still need to measure crucial metabolites linked with n-6 and n-3 PUFAs such as Arachidonic acid, Linoleic acid etc and its products. 

3.) Although it might be out of the scope of this study, discussion should include some points about how in the future such diet studies can be used to evaluate other inflammaging parameters including motor functions and cardiometabolic health. 

Author Response

Review 1

We greatly appreciate the time spent on our manuscript revision. All of the comments motivated us to re-evaluate our outcomes in order to deliver an improved manuscript.

Although the authors implemented some of the suggested edits, the authors still need to address some of these questions.

  1. The authors still need to update the abstract and introduction section and mention the studies were carried out in a subset of population belonging to the same geographic location and having mostly similar lifestyle and diet.

Thank you for your suggestion. The sections Abstract and 1. Introduction have been complemented with the following information “…the Polish elderly population”, and 2.1. Participants “…were recruited from the University of the Third Age in Zielona Gora”. Section 6. Limitations has been completed with the following excerpt “…confined predominantly to population belonging to the same geographic location and having mostly similar lifestyle.”

Indeed, the participants of our study have shown the same interests or lifestyle. They have been the students of the University of the Third Age (U3A), which is an organization encouraging the elderly over 65 years of age to stay cognitively or physically active (by participating in many educational programmes, including arts, classical studies, discussion classes, computer courses, crafts, drama, film/cinema studies, history, languages, literature, music, social sciences, and physical activity Tai-Chi, Nordic walking, Swimming etc.). They have also represented a generation with low consumption of vegetables, fruits and fish, which we could observed in our study.

  1. The authors still need to measure crucial metabolites linked with n-6 and n-3 PUFAs such as arachidonic acid, linoleic acid etc. and its products.

Thank you for pointing this out. The results of measurements of crucial metabolites linked with n-6 and n-3 PUFA such as arachidonic acid, linoleic acid, EPA and DHA have been presented in Table 3. Sections 3.2. Diet analysis (Line 231-234) and 4. Discussion (Line 400-407) have been revised accordingly.

  1. Although it might be out of the scope of this study, discussion should include some points about how in the future such diet studies can be used to evaluate other inflammaging parameters including motor functions and cardiometabolic health.  

Section 4. Discussion has been enriched with information concerning cardiometabolic health and motor functions.

Line 400-407: Recently, Felix-Soriano et al. [Nutrients 2021] demonstrated that supplementation with a DHA-rich fish oil concentrate had beneficial effects on cardiometabolic health markers in postmenopausal women. In our study, the adherence to dietary recommendations concerning PUFA was better in a diet of the elderly in LGI compared to HGI group. The dietary content of n-6 PUFA such as linoleic and arachidonic acid were predominant in a diet of HGI group. There is evidence that a high n-6 fatty acid diet inhibits the anti-inflammatory and inflammation-resolving effect of the n-3 fatty acids [Innes and Calder Prostaglandins Leukot Essent Fatty Acids 2018].

Line 443-448: Decreased plasma albumin have been considered as a risk factor for nutritional deficiencies and frailty in the elderly [Alberro et al Sci Rep 2021, Onem et al. [Arch Gerontol Geriatr 2010]. Onem et al. [Arch Gerontol Geriatr 2010] showed the relationships between albumin concentration and cognitive and motor functions in patients from nursing homes. They concluded that restoring the reference values of albumin could improve cognitive and motor functions in the elderly.

Reviewer 2 Report

The authors in this manuscript investigated the association between anti-inflammatory diet, lifestyle exercise, and the inflammatory profile in the elderly. The study is well designed and measured many relevant variables. The authors addressed comments from previous reviewers. However, the confounding effects in the LGI and HGI comparison analysis are concerned.

The authors showed the female vs male body composition in table 1, but there was not a body-composition comparison between LGI and HGI groups. If body composition variables are significantly different, authors need to consider those confounding effects in the statistical analysis. 

For example, the authors showed a significant difference in gait speed between LGI and HGI. If age were also significantly different between these two groups, the gait speed difference might result from the age difference. Thus, authors need to consider adding multivariate regression analysis for the comparison to control the confounding effects. This confounding effect concern may exist in all LGI and HGI comparisons.

However, if there is no significant body composition difference between LGI and HGI groups, authors do not need to add any more analysis and only need to add one table to show the body composition information between LGI and HGI groups.

Here are some minor points:

1 line 199

Authors may consider replacing Rstudio version information with R version information.

2 line 205

The eta-square analysis seems not to be an R base function. Authors need to specify what package is used. Also, the authors defined four levels of effect size and may consider adding one reference.

3 line270

The sentence is too conclusive. Current results cannot clearly show inflammatory process impairs physical performance.

4 line 278

Leucocytes and granulocytes are two different levels terms. Since leucocytes are composed of granulocytes and lymphocytes, authors may consider rewriting this sentence.

Author Response

Review 2

We greatly appreciate the time spent on our manuscript revision. All of the comments motivated us to re-evaluate our outcomes in order to deliver an improved manuscript.

The authors showed the female vs male body composition in table 1, but there was not a body-composition comparison between LGI and HGI groups. If body composition variables are significantly different, authors need to consider those confounding effects in the statistical analysis. For example, the authors showed a significant difference in gait speed between LGI and HGI. If age were also significantly different between these two groups, the gait speed difference might result from the age difference. Thus, authors need to consider adding multivariate regression analysis for the comparison to control the confounding effects. This confounding effect concern may exist in all LGI and HGI comparisons. However, if there is no significant body composition difference between LGI and HGI groups, authors do not need to add any more analysis and only need to add one table to show the body composition information between LGI and HGI groups.

Thank you for pointing this out. The results of body-composition comparison between LGI and HGI groups have been presented in Table 2. Section 3.1. Body composition has been revised accordingly.

Here are some minor points:

1 line 199. Authors may consider replacing Rstudio version information with R version information.

The section 2.9. Statistical analysis has been corrected: “Statistical analyses were performed using the R version 4.0.3 [30].”

 2 line 205. The eta-square analysis seems not to be an R base function. Authors need to specify what package is used. Also, the authors defined four levels of effect size and may consider adding one reference.

We absolutly agree with Reviewer. Eta-square analysis is not available in the R base installation. However, it is available in optional packages, such as DescTools (the EtaSq function) or effect size (the eta_squared function). We used the first package in our analysis.

The section 2.9. Statistical analysis has been rewritten: Additionally, eta-squared (η2) based on EtaSq function from DescTools R package was used to measure the effect size which is indicated as having no effect if 0≤ η2< 0.01, a small effect if 0.01≤ η2< 0.06, a moderate effect if 0.06≤ η2< 0.14, and a large effect if η2 ³0.14.”

3 line 270. The sentence is too conclusive. Current results cannot clearly show inflammatory process impairs physical performance.

Following the Reviewer’s suggestion, the sentence has been removed.

4 line 278. Leucocytes and granulocytes are two different levels terms. Since leucocytes are composed of granulocytes and lymphocytes, authors may consider rewriting this sentence.

The sentence has been rewritten: “The white blood cell count fell within the referential range in all participants whereas higher granulocytes and GRA% were detected in HGI compared to LGI group.”

Round 2

Reviewer 2 Report

The authors have clearly addressed my concerns. Congrats. 

This manuscript is a resubmission of an earlier submission. The following is a list of the peer review reports and author responses from that submission.

Round 1

Reviewer 1 Report

This manuscript describes a study of aged adults in Poland and evaluates for associations between dietary intake of anti-inflammatory compounds and physical status. Overall, the study is well designed, but improvements in the description could be made. The manuscript is very long and the reader would appreciate more concise delivery of the results/discussion.  What would help in this aspect is the removal of discussion items that refer to associations that are not significant.  There are likely some very interesting findings, but these are greatly diluted among the highlights of “trending” associations. In addition, the tables could greatly be improved through reductions in the use of abbreviations and inclusion of statistics.  This is especially relevant in this version of the manuscript where so many discussed findings are not statistically significant. The manuscript does require significant revisions for language and length, but does have the potential to make a contribution to the field.

Major comments

  1. The introduction requires a clearer statement on the definition of inflammation and the relationship to dietary intake.
  2. Further details of participant recruitment are needed. How were participants identified/recruited? Why were medical records not evaluated for inclusion/exclusion criteria?
  3. No mention of an assessment of cognitive function was used in the inclusion/exclusion criteria which is concerning as dietary intake was solely assessed by recall and the participants are of advanced age where cognitive decline is highly prevalent.
  4. Results lines 210-228: The authors report changes in diets between high and low groups which do not meet statistical significance. Just because one group had a higher average than the other, does not alone justify it as a noteworthy finding.
  5. Line 223: “the diet can enhance inflammaging.” The authors are only stating an association not a cause and effect relationship. Please revise or delete.
  6. Table 3: Please include statistics. Why are not average ± SD values provided? This would help the reader understand how variable the intake of these compounds are.
  7. Section 3.3 Physical Performance: Data for the associations described in this section are not provided – please include a table with this information.
  8. Discussion: Several references to “significant” findings that show no statistical significance in the results. For example, Lines 354-355 – This statement is not supported by the data as the associations between HCI/LCI group status and total carbohydrates, glycemic index and glycemic load are not statistically significant.

Minor comments

  1. Lines 82-83: Please clarify “the major feature of the aged immune system” – just state this feature?
  2. Lines 90-91: “lifestyle of the participants were controlled by using” – requires clarification as the investigators are not able to control participants lifestyle
  3. Line 92: Please clarify what “same access to medical healthcare” means. This is needed since access to healthcare is highly variable in most populations.
  4. Lines 122-123: “recurrence of measurement was 98%” Please clarify. Were the multiple measurements taken exactly the same or was there a threshold where measurement error was acceptable? Could be represented as the average difference of the 2 measurements ± standard deviation?
  5. Please revise “old elderly”, “old elderly adults” and “old elderly senior” etc – this is very redundant language. Also be consistent throughout manuscript.
  6. Lines 218-221: Please include statistics in statement
  7. Line 221: “large influence” please provide more specific language regarding interpretation of statistic.
  8. Tables 2, 3, 4, 5: for clarity, please reduce the usage of abbreviations. For example, many of the substances can be spelled out as can the
  9. Line 308: “strong influence” please provide more specific language regarding interpretation of the statistic.
  10. Line 351: “In both groups” please identify/be more specific as no mention of the groups studied had been introduced in the discussion.
  11. Line 353: “appeared less varied” what does this mean – can you be more specific/quantitiative?

Reviewer 2 Report

Dear Authors,

the manuscript deals with several interesting concepts on the relationship between inflammation, diet and physical activity in the elderly. 

Although a single 24-hour dietary recall is recognized as one of the limitations of the study together with the sample size (very low), it is difficult to draw scientifically valid conclusions from the results obtained using this method (multiple calls or other tools are needed to avoid important bias - Sven Knüppel, Kristina Norman, Heiner Boeing, Is a Single 24-hour Dietary Recall per Person Sufficient to Estimate the Population Distribution of Usual Dietary Intake?, The Journal of Nutrition, Volume 149, Issue 9, September 2019, Pages 1491–1492, https://doi.org/10.1093/jn/nxz118).

Furthermore, the various sections of the manuscript are confusing: the introduction is very verbose, in the methods section there are already data that should go into the results, in the results section there are information that should go into the discussion, the subparagraphs of the methods and results should be rearranged and reordered, the results alternate samples of 60 people and 41 people (comparing them as if they were the same thing), the p-value (significance) is not present in the paragraph on physical activity, etc..

Finally, the tables (especially Tables 1, 3 and 4) and the figures (in particular the Figure 1) are incorrect or not easily understood.

Given all these observations, I think that important changes are needed before proceeding with the submission of this article. I am confident that, once the manuscript is modified, it could be very interesting given the topics covered.

Reviewer 3 Report

The authors here hypothesizes that there is a link between diet and lifestyle on inflammatory profile in the elderly populations who are specifically from Poland. Here the authors explain in detail the types of diets and the following inflammatory patterns/ biomarker changes in those individuals. The story has some gaps that need to be addressed, which are as a follows:

  1. Since, the studies were carried out on a few elderly individuals who are from the same geographic location, have similar diets, lifestyles etc. the title, abstract and introduction can be very misleading and is an over generalization. Hence, the title needs to be updated and needs to mention it is from a specific region and has small sample size. Similarly, the abstract/ Introduction needs to be updated too.
  2. The introduction also needs to be updated with information about the molecular pathways that are linked with inflammaging including mTOR, Insulin/ IGF signaling, Endocannabinoid system, NRF2 pathway etc.
  3. Since the study included individuals ranging from ages 65 to 80. A graph showing comparisons between the individuals in this study belonging to higher age group and lower age group and comparison of key biomarkers in these subgroups would be helpful and informational to the readers. Additionally, a graph showing the age group distribution would be also helpful.
  4. Also, some other inflammaging biomarkers also need to be included in this study which include metabolites such as Lactate (PMID: 31708446)
  5. Additionally, since the diet contains n-6 and n-3 PUFA, it will also be important to measure Arachidonic acid and its derivatives such as Endocannabinoids and other pro-inflammatory molecules which are crucial biomarkers of the process.
  6. Readers could benefit from a summary figure at the end which shows the overall hypothesis and crucial results.